# Dissecting the Crosstalk between Endothelial Mitochondrial Damage, Vascular Inflammation, and Neurodegeneration in Cerebral Amyloid Angiopathy and Alzheimer’s Disease

**DOI:** 10.3390/cells10112903

**Published:** 2021-10-27

**Authors:** Rebecca M. Parodi-Rullán, Sabzali Javadov, Silvia Fossati

**Affiliations:** 1Alzheimer’s Center at Temple, Lewis Katz School of Medicine, Temple University, Philadelphia, PA 19140, USA; rebecca.parodi-rullan@temple.edu; 2Department of Physiology, University of Puerto Rico School of Medicine, San Juan, PR 00921, USA; sabzali.javadov@upr.edu

**Keywords:** mitochondria, Alzheimer’s disease, cerebral amyloid angiopathy, inflammation, neurodegeneration, amyloid, endothelial cells

## Abstract

Alzheimer’s disease (AD) is the most prevalent cause of dementia and is pathologically characterized by the presence of parenchymal senile plaques composed of amyloid β (Aβ) and intraneuronal neurofibrillary tangles of hyperphosphorylated tau protein. The accumulation of Aβ also occurs within the cerebral vasculature in over 80% of AD patients and in non-demented individuals, a condition called cerebral amyloid angiopathy (CAA). The development of CAA is associated with neurovascular dysfunction, blood–brain barrier (BBB) leakage, and persistent vascular- and neuro-inflammation, eventually leading to neurodegeneration. Although pathologically AD and CAA are well characterized diseases, the chronology of molecular changes that lead to their development is still unclear. Substantial evidence demonstrates defects in mitochondrial function in various cells of the neurovascular unit as well as in the brain parenchyma during the early stages of AD and CAA. Dysfunctional mitochondria release danger-associated molecular patterns (DAMPs) that activate a wide range of inflammatory pathways. In this review, we gather evidence to postulate a crucial role of the mitochondria, specifically of cerebral endothelial cells, as sensors and initiators of Aβ-induced vascular inflammation. The activated vasculature recruits circulating immune cells into the brain parenchyma, leading to the development of neuroinflammation and neurodegeneration in AD and CAA.

## 1. Introduction

With an increase in life expectancy, there has been an increase in the incidence of age-related diseases, particularly Alzheimer’s disease (AD) which is the most prevalent cause of dementia, representing around 60–80% of all dementia cases. Approximately 6.2 million Americans are currently living with AD, and due to the increase in the aging population, it is expected that this number will grow to 13.8 million by 2060 [1]. Currently, there is no effective therapy for AD. In 2021, after nearly twenty years of attempts to develop new therapeutic strategies, aducanumab, a human monoclonal antibody that selectively targets aggregated amyloid-β (Aβ), has been approved by the FDA (through the accelerated approval pathway) for treatment of patients with AD [2]. However, the therapeutic effectiveness of aducanumab is still debated by many research groups [2,3,4]. The high prevalence of AD among aged individuals and lack of effective therapy denotes the importance of developing new therapeutics, and to do so we need to have a better understanding of the complexity of the disease.

Clinical manifestations of AD include progressive cognitive impairment, which develops after years of asymptomatic pathological and molecular changes. Pathologically, AD is characterized by the presence of extracellular senile plaques of aggregated Aβ and intraneuronal neurofibrillary tangles of hyperphosphorylated tau protein [5]. AD is a multifactorial disease and vascular dysfunction is one of the most common co-pathologies that may also be an important contributor to disease progression [6,7]. Indeed, in over 80% of AD patients, and to a smaller extent in non-demented individuals, Aβ is also found around cerebral vessels, a condition known as cerebral amyloid angiopathy (CAA) [8]. The vascular accumulation of Aβ occurs mostly in cortical and leptomeningeal arteries and capillaries and is often associated with cerebral microhemorrhages, increasing blood–brain barrier (BBB) permeability, and inflammation [9,10]. However, it is important to note that CAA is not exclusive to amyloidosis due to Aβ [11,12].

The cerebrovascular accumulation of Aβ results in neurovascular dysfunction, neuroinflammation, and progression of the neurodegenerative process. Elevated levels of IL-6, TNFα, and IL-1β have been detected in animal models of amyloidosis, such as Tg2576 and 3xTg mice, and in the cerebrospinal fluid (CSF) of AD patients [13,14], indicating the presence of sustained inflammation [15] that accompanies AD pathology. In particular, cerebrovascular inflammation has been shown in animal models of amyloidosis as well as in post-mortem human brains [16,17]. These perivascular inflammatory processes may precede and stimulate parenchymal plaque deposition [17]. Indeed, increasing evidence suggests that vascular dysfunction can occur before classical AD pathology [18,19,20,21]. Vascular inflammation leads to immune cell extravasation into the brain parenchyma and widespread neuroinflammation, which has been shown to positively correlate with the levels of Aβ in mouse models of amyloidosis and in AD patients [22,23,24]. Altogether, recent studies indicate the importance of understanding key vascular events that contribute to the development of neuroinflammation and neurodegeneration.

Amongst the vascular mechanisms that may contribute to neurodegeneration, alterations in the metabolism and function of cerebrovascular mitochondria play a crucial role in Aβ-induced dysfunction of endothelial cells (EC), BBB permeability, and inflammation [25,26,27,28]. Therefore, the preservation of mitochondrial function is an attractive strategy towards the prevention of age-associated neurodegenerative diseases such as AD and CAA [29]. In this review, we summarize and discuss previous studies investigating the role of cerebrovascular mitochondria, particularly EC mitochondria, as mediators of neuroinflammation in CAA and AD. Additionally, we will focus on the role of Aβ-induced mitochondrial dysfunction in ECs, how this may initiate the inflammatory process in the neurovascular unit, and discuss the pathways that may lead to neurodegeneration.

## 2. Endothelial Cells and the Neurovascular Unit

The neurovascular unit (NVU) is composed of ECs lining the vascular lumen, smooth muscle cells (SMC) present in the arteries and veins, pericytes, and astrocytes. Cerebrovascular cells are also connected with other brain cells such as neurons and microglia and thus, the NVU provides the structural and functional relationship between the brain and cerebral vasculature. Particularly, it participates in the maintenance and regulation of cerebral circulation, vascular permeability through the BBB, and mediation of brain inflammatory pathways. Therefore, NVU dysfunction is a prominent and early feature in the AD brain [30,31,32,33,34]. The BBB serves as a barrier for the movement of molecules from the blood to the cerebral parenchyma. Its unique structure permits the regulation of the passage of molecules such as peptides, toxins, nutrients, and metabolites between the blood and the brain. This function of the BBB is primarily mediated by cerebrovascular ECs, which differ from other ECs due to the lack of fenestrations, low pinocytic activity, the expression of tight junctions (TJs, one of the characteristics of the BBB), and high mitochondrial content [35,36]. Indeed, the high mitochondrial mass in comparison to peripheral ECs indicates the significant role played by the mitochondria in the metabolism and function of cerebral ECs [37]. Notably, mitochondrial content was found to be reduced in the brains of patients with AD [38]. Mitochondria have been shown to play a crucial role in the downstream pathways mediating EC damage during AD and CAA [39], vascular inflammation, and neurodegeneration [40,41,42,43,44].

## 3. Failure of Aβ Clearance and CAA Development

Sequential cleavage of the amyloid precursor protein (APP) by β-secretase and γ-secretase generates Aβ peptides, which are released from neurons and other brain cells into the brain parenchyma. Aβ peptides aggregate to form oligomers, fibrils, and eventually the amyloid plaques observed in AD brains. The main Aβ peptides found in the brain parenchyma and around the cerebral vessels are Aβ40 and Aβ42, although the processing of APP leads to the generation of multiple other Aβ fragments varying in length and aggregation profile [28,45]. It is speculated that the accumulation of Aβ in the brain and around cerebral vessels may be due to increased APP cleavage, Aβ aggregation, as well as decreased Aβ clearance, or combinations of these mechanisms. Due to the lack of conventional lymphatic vessels in the brain, clearance of Aβ from the parenchyma into the lymphatic system is thought to occur through intramural periarterial drainage (IPAD) [46,47,48] and/or through the glymphatic system [49,50,51]. Clearance of metabolites through IPAD occurs primarily along the basement membranes of cerebral capillaries and arteries. The precise mechanism of the movement of molecules and peptides, including Aβ, along the basement membrane of the cerebral vasculature, remains unclear. Several mechanisms such as arterial pulsations and vascular smooth muscle cell contractility have been proposed [52,53]. The glymphatic system hypothesis is based on the observation that tracers injected into the subarachnoid CSF readily flow into the brain along the outside of the penetrating blood vessels. This suggests that CSF flows into the brain tissue along para-arterial spaces and exits via a para-venous route, aided by astrocytic end-feet through aquaporin 4 [49,54]. The meningeal lymphatic vessels might provide another important clearance route for brain products, which is recently gaining relevance in neurodegenerative disorders such as AD [55,56].

Thus, multiple routes participate in cerebral clearance of waste products including Aβ. These pathways may act in parallel or in a complementary fashion to maintain brain homeostasis [57]. Most importantly, failure of Aβ clearance by one or multiple mechanisms may lead to Aβ deposition along the cerebral vasculature (CAA), leading to NVU dysfunction. The accumulation of Aβ around the cerebral vasculature occurs in a progressive fashion: first, Aβ surrounds the vessels in the adventitia and tunica media, provoking vascular SMC damage and eventually death. Later, as the accumulation progresses, the tunica media is replaced by Aβ fibrils which later may lead to EC death, BBB breakdown, and microhemorrhages [58]. Indeed, early studies report that in human AD brains, the accumulation of Aβ fibrils in cerebral vessels provokes EC degeneration and reduction in vessel size [59]. This vascular dysfunction, in turn, is associated with cerebral hypoperfusion, neuroinflammation, parenchymal Aβ accumulation, plaque formation, and AD progression. Therefore, it is important to understand the effects of Aβ accumulation on the cerebral vasculature and its role in the progression of the neurodegenerative process in the brain of patients with AD and CAA.

## 4. Activation of Endothelial Inflammatory Pathways by Aβ

Inflammatory pathways may be activated through pattern recognition receptors (PRRs) such as membrane-bound toll-like receptors (TLRs) and C-type lectin receptors (CLRs), as well as cytoplasmic NOD-like receptors (NLRs) and retinoic acid inducible gene I (RIG-I)-like receptors (RLRs). PRRs are expressed by immune and non-immune cells such as fibroblasts, epithelial cells, and ECs. They are responsible for the detection of pathogen-associated molecular patterns (PAMPs) and danger-associated molecular patterns (DAMPs). The PAMPs are foreign molecules derived from viruses and bacteria whereas DAMPs are danger signals originating from the organism and are associated with cellular damage [60]. In addition to the PRRs, DAMPs can also activate non-PRR such as the receptor for advanced glycation end products (RAGE) and triggering receptors expressed on myeloid cells (TREMs) [60]. The activation of PRRs and non-PRRs by DAMPs is also referred to as sterile inflammation, due to the absence of pathogens. Interestingly, many of the DAMPs associated with sterile inflammation are of mitochondrial origin, highlighting the crucial role of mitochondria in neuroinflammation [26].

In ECs, Aβ transport across the plasma membrane is regulated by the low-density lipoprotein receptor-related protein 1 (LRP1) to deliver the peptide from the brain parenchyma to the blood. LRP1-mediated endocytosis regulates cellular Aβ uptake by binding to Aβ either directly or indirectly through its co-receptors or ligands. Transport of Aβ from the periphery to the brain is mediated by RAGE [61,62]. RAGE is a receptor of the immunoglobulin superfamily that is found on the luminal side of ECs of the BBB. RAGE-mediated transport of Aβ across the BBB may lead to Aβ deposition in the brain parenchyma and, in addition, induce inflammatory responses [62]. The pro-inflammatory effects of the Aβ–RAGE interaction have been extensively studied in various cell and animal models [62,63,64].

Activation of inflammatory responses by ECs is characterized by the expression of cellular adhesion molecules (CAMs), such as vascular cell adhesion molecule-1 (VCAM-1), intercellular adhesion molecule-1 (ICAM-1), and E-selectin as well as disruption of the BBB, and release of pro-inflammatory cytokines. The presence of DAMPs has been shown to induce a pro-inflammatory state in ECs that leads to the recruitment of immune cells, triggering the further release of pro-inflammatory cytokines and activation of resident microglial cells [24]. One of the most important mediators of the inflammatory state in ECs is the NLR family pyrin domain containing 3 (NLRP3). Elevated levels of NLRP3 have been observed in human AD brains [65] and animal models of amyloidosis [66]. Cytosolic NLRP3 can be activated indirectly through its capabilities of sensing intracellular danger associated molecules, such as mitochondrial DAMPs, or directly by the endothelial CD36 receptor upon Aβ binding [67,68]. Activation of NLRP3 and the recruitment of the adaptor protein, apoptosis-associated speck-like protein containing a caspase recruitment domain (ASC), induces the catalytic cleavage of pro-caspase-1 to active caspase-1. Active caspase-1, in turn, cleaves pro-inflammatory cytokines pro-IL-1β, pro-IL-18, and gasmerdin D, producing their active forms. Gasmerdin D forms non-selective pores in the plasma membrane to facilitate the release of cytokines, and thus activates pyroptosis, a form of inflammatory cell death [60,69,70]. It has been suggested that NLRP3 can also activate the nuclear factor kappa-light-chain-enhancer of activated B cells (NFκB) which, in turn, increases the transcription of pro-inflammatory cytokines [71] and thereby initiates a positive feedback loop to further stimulate the inflammatory response. Interestingly, NFκB is also capable of downregulating the endothelial nitric oxide synthase (eNOS), thereby decreasing NO bioavailability and vascular relaxation [72].

## 5. Pathological Consequences of Aβ on Endothelial Mitochondria

Cerebrovascular ECs are the gatekeepers of brain health through the maintenance of the BBB and regulation of movement of ions and molecules across the blood–brain interface. Due to their important role in brain cell health, alterations of mitochondrial structure and function in cerebrovascular ECs induce cell dysfunction, loss of BBB integrity, and inflammation, eventually leading to cell death [27,39,73,74,75]. Cerebrovascular ECs contain more mitochondria than any other EC in the body. Mitochondria account for 8–11% of cytoplasmic volume in brain ECs, compared to only 2–5% in non-cerebrovascular ECs [35,37].

In CAA, Aβ can induce EC apoptosis through the direct binding and activation of the TRAIL death receptors (DR) DR4 and DR5 [76]. These effects appear to be preferentially mediated by oligomeric and/or protofibrillar amyloid species, while species that have fast aggregation dynamics and quickly form fibrils, as well as Aβ peptides that remain mostly monomeric, fail to induce EC apoptosis. Fibrillar species, however, are involved in other EC damage pathways, such as increases of BBB permeability [28]. The activation of the TRAIL DRs by Aβ oligomers and protofibrils triggers the extrinsic apoptotic pathway, leading to cleavage of caspase 8 to its active form, cleavage of BH3-interacting domain death agonist (BID), and release of cytochrome C (CytC) from mitochondria with subsequent activation of the intrinsic apoptotic pathway. Indeed, loss of mitochondrial membrane potential, a key event preceding CytC release, was observed in the presence of Aβ, along with the release of reactive oxygen species (ROS) [27,39,77,78,79]. Interestingly, the CD36 receptor appears to be involved in the progression of CAA and loss of TJ proteins in Tg2576 mice, a model characterized by amyloid parenchymal deposition as well as CAA, through NADPH oxidase-mediated increases in ROS [80,81], however the contribution of CD36 to Aβ-induced mitochondrial dysfunction in ECs is unclear.

Additionally, in animal models of amyloidosis, it has been shown that the mitochondrial permeability transition pore (mPTP) is an important contributor to mitochondrial dysfunction, neuronal death, and cognitive dysfunction in AD models [82], but only a few studies have investigated the potential effects of mPTP in mediating the detrimental effects Aβ on cerebrovascular ECs [83]. In the following sections, we will address the possible contribution of the mPTP, mitochondrial ROS (mtROS), and mitochondrial DNA (mtDNA) to inflammation during Aβ pathology.

## 6. Mitochondrial DAMPs as Initiators of Inflammation

The mitochondrion is thought to have originated from an endosymbiotic uptake of an α-proteobacteria [84]. The immune system serves as a protective wall against cellular damage and pathogens; therefore, it is of no surprise that mitochondrial fragments may trigger an immune response. In turn, activators of inflammatory pathways also trigger mitochondrial damage such as mtROS production and loss of mitochondrial membrane potential [85], potentially resulting in a vicious cycle between mitochondrial dysfunction and inflammation. In addition, inflammatory mediators could also amplify mitochondrial dysfunction, as NLRP3-mediated Caspase-1 activation was shown to induce mtROS production, loss of mitochondrial membrane potential, and mitochondrial membrane permeabilization, resulting in the release of mitochondrial DAMPs in bone-marrow-derived macrophages (BMDM) [86]. Several mitochondrial components have been identified as DAMPs (Figure 1). In this section, we will attempt to highlight the part they play in innate immunity and whether their role has been identified in cerebral ECs in the context of AD and CAA.

### 6.1. Mitochondrial ROS

Accumulation of ROS in cells can occur due to increased ROS production in the cytosol (NAD(P)H oxidases) and mitochondria (electron transport chain (ETC), tricarboxylic acid (TCA) cycle, monoamine oxidases, etc.), and/or decreased activity of antioxidant enzymes (superoxide dismutase (SOD), catalase, peroxidase, etc.) and systems (glutathione, tocopherol, thioredoxin, ascorbic acid, etc.). Mitochondria are the major source of ROS, with over 10 different sites of ROS production [87]. In the inner mitochondrial membrane (IMM), electrons are moved across the ETC complexes accompanied by oxygen consumption. The proton gradient generated by ETC across the IMM stimulates ATP production through oxidative phosphorylation (OXPHOS). Mitochondrial ROS production increases in dysfunctional mitochondria as a result of downregulation of ETC and OXPHOS. Decreased activity of ETC complexes, particularly complexes I and III, stimulates excessive mtROS production due to increased electron leakage by these complexes generating superoxide anion (O_2_^•−^) [88]. The superoxide anion is converted to H_2_O_2_ and produces other oxygen radicals through Fenton reactions that are collectively known as ROS. Although production of mtROS occurs under normal physiological conditions, it is normally balanced by the antioxidant defense that includes SOD, catalase, peroxidase, thioredoxin, and glutathione systems.

Increased generation of ROS has been implicated in several diseases, including AD and CAA. Aβ has been shown to increase mtROS production in neurons [27,78] and cerebral ECs [27]. High ROS levels were observed in Tg2576 mice [89], along with a reduction in cerebral blood flow (CBF) [90]. Likewise, high ROS levels were found in the cerebral vasculature of the 3xTg mice, that develop amyloidosis and tauopathy [13]. In a mouse model expressing the human APP with the Swedish and Indiana mutations [91], downregulation of SOD2, a mitochondrial-matrix localized SOD, promoted vascular amyloidosis but not parenchymal amyloid deposition [43]. In contrast, SOD2 overexpression, as expected, was able to increase CBF, thereby reducing vascular dysfunction in Tg2576 mice [90]. Altogether, these studies demonstrate a crucial role of mtROS in the development of AD and CAA. Although the generation of ROS can be enhanced by upregulation of non-mitochondrial (cytoplasmic) sites such as NADPH oxidases, here, we will focus on the role of mtROS in vascular inflammation and neurodegeneration.

Substantial evidence has stipulated that Aβ can induce mtROS accumulation and EC inflammation [13,27,43,78], however, the mechanism underlying the effects of Aβ on cerebral EC mitochondria remains unclear. Aβ may induce mtROS generation as a result of inhibition of ETC complexes [78], by modulation of downstream pathways after its binding to cell surface receptors, or through downregulation of antioxidant systems [43]. As mentioned above, Aβ has been shown to bind to and activate RAGE [92], leading to increased RAGE expression [64,93], ROS generation, and activation of the transcription factor NFκB [43,94,95,96], a key regulator of pro-IL-1β, pro-IL18, and NLRP3 expression [25].

ROS are involved in NLRP3-mediated neuroinflammation induced by oligomeric Aβ in microglia [97], and polychlorinated biphenyls (PCB) 118-induced pyroptosis in ECs [98]. In rat brain ECs, an increase of superoxide induced the release of TNFα and Aβ [15]. Similarly, ROS production was associated with the release of vascular IL-6, which was attenuated by the presence of a ROS scavenger in 3xTg mice [13], highlighting the role of ROS as mediators of vascular inflammation. Additionally, in BMDMs, attenuation of mtROS with mito-TEMPO, a mitochondria-targeted antioxidant, reduced caspase-1 activation and the release of IL-1β and IL18 in response to lipopolysaccharide (LPS) and ATP; two activators of the NLRP3 inflammasome [99]. Thus, accumulating evidence suggests that excessive ROS in ECs induces the activation of an inflammatory response and the NLRP3 inflammasome plays an important role in mediating the effects of ROS. In the context of CAA and AD, Aβ may be the culprit of such an increase in mtROS generation.

### 6.2. Mitochondrial Permeability Transition Pore

The mPTP is a pathological pore that forms across the IMM in response to various pathological stimuli associated with matrix Ca^2+^ overload, elevated mtROS, and loss of mitochondrial membrane potential. The mPTP is a non-selective pore that allows the passage of solutes up to 1.5 kDa across the IMM. Under physiological conditions, the IMM is impermeable and the transport of ions and other solutes through the membrane is regulated by membrane transporters and other exchange mechanisms. Therefore, the formation of the mPTP causes mitochondrial swelling due to increased colloid osmotic pressure in the mitochondrial matrix which, in turn, leads to rupture of the outer mitochondrial membrane (OMM). As a result, apoptotic proteins such as CytC are released from the intermembrane space of mitochondria to the cytosol. Although the formation of the mPTP is well documented and has been observed under electrophysiological preparations [100], the molecular identity of the pore is unknown. Ironically, inhibition of the mPTP would be an ideal target for various diseases including myocardial infarction [101,102], cerebral ischemia [103,104], and AD [105,106,107]. Although the molecular identity of the mPTP is unknown, the peptidyl-prolyl *cis*-*trans* isomerase Cyclophilin D (CypD) has been accepted as a major mPTP regulator [108]. Cyclophilin D, localized in the mitochondrial matrix, was found to increase in AD-affected brain regions. Aβ binds to CypD, and the Aβ-CypD complex was detected in Aβ-rich mitochondria from AD brain and transgenic AD mice [44,82]. These studies found that CypD deficiency prevented Aβ-mediated mitochondrial and synaptic dysfunction, suggesting that the effects of Aβ to induce mPTP opening are mediated through its interaction with CypD (Aβ binding partner). The role of the mPTP in cerebral EC mitochondrial dysfunction has not been fully studied in the context of CAA and AD. However, we will attempt to highlight its possible role in mediating mitochondrial dysfunction and inflammation.

Several studies have demonstrated that cyclosporin A (CsA), an inhibitor of CypD, protects against inflammation. In BMDMs, treatment with CsA inhibited caspase-1 release and IL-1β secretion after LPS and ATP treatment [99,109,110]. However, *CypD*^−/−^ macrophages still exhibited NLRP3 activation [109] questioning whether the mPTP opening is involved in NLRP3 activation. It should be pointed out that CsA is not a specific CypD inhibitor, as it also targets calcineurin, a serine/threonine protein phosphatase in the cytosol [111], thereby suggesting that the effects of CsA on inflammasome activation are mPTP-independent. Further studies are required to establish the role of the mPTP in NLRP3 and inflammasome activation by using more specific CypD inhibitors such as sanglifehrin A (SfA) [111]. It is also important to consider that none of these drugs completely inhibit mPTP activation; CypD is an important mPTP regulator but not an essential component, as *CypD^−/−^* cells are still able to form the mPTP [112]. Therefore, current data does not confirm or exclude the participation of the mPTP in NLRP3 activation. In addition, the use of direct (CypD independent) mPTP inhibitors [113,114] could also shed a light on the role of the mPTP in Aβ-mediated mitochondrial dysfunction and NLRP3 inflammasome activation in cerebral ECs during CAA and AD pathology.

One of the key inducers of mPTP formation is an increase in mtROS [115,116]. This phenomenon is commonly known as “ROS induced ROS release” [87], when mPTP-induced ROS in one mitochondrion triggers mPTP opening in adjacent mitochondria with subsequent ROS release. Therefore, the mPTP would be a viable mechanism for Aβ-induced mtROS release in cerebral ECs, which may lead to the activation of inflammation in the neurovascular unit. Interestingly, the formation of the mPTP has been suggested to mediate the release of mtDNA from mitochondria into the cytoplasm [117,118]. Cytoplasmic mtDNA has been extensively recognized as a DAMP and mediator of the inflammatory response in other pathologies [26,119]. Overall, the mPTP opening in cerebral ECs may be involved in the Aβ-induced release of mitochondrial DAMPs (mtROS, mtDNA), which can result in inflammation.

### 6.3. Mitochondrial DNA

The double stranded mtDNA lacks exons and introns, is circular, packaged into nucleoids instead of histones [120], and contains hypomethylated CpG nucleotides, making mtDNA share more similarities to bacterial DNA than to nuclear DNA. Indeed, one of the first studies by Collins et al. demonstrated that injection of mtDNA, but not nuclear DNA, into mice joints lead to inflammation, increased activity of NFκB, and elevated TNFα production [121]. Since then, an increasing amount of literature has confirmed the inflammatory nature of mtDNA and identified several pathways involved in mtDNA-induced inflammatory responses [119,122].

Cytoplasmic double stranded DNA (dsDNA) is recognized by cyclic GMP-AMP synthase (cGAS). This enzyme recognizes dsDNA in a sequence-independent manner. Therefore, it can recognize both pathogen and host dsDNA, such as the mtDNA. Once activated by dsDNA, cGAS generates cyclic GMP-AMP (cGAMP) from ATP and GTP, which then activates the stimulator of interferon genes (STING) on the endoplasmic reticulum (ER) membrane [123]. Activated STING induces the phosphorylation and activation of the transcription factors interferon regulatory factor 3 (IRF3) and NFκB [118,124,125], among other mediators of inflammation. Confirming this mechanism, studies have shown that in retinal [118] and lung [126] microvascular ECs, the release of mtDNA to the cytoplasm resulted in the activation of the cGAS-STING pathway and pro-inflammatory transcription factors IRF3 and NFκB. Additionally, exogenous administration of mtDNA to retinal microvascular ECs induced the activation of the cGAS-STING pathway [118]. Thus, many studies confirm that cytosolic mtDNA is capable of inducing an inflammatory response, however, the mechanisms of its release from mitochondria into the cytoplasm have yet to be fully elucidated.

Recent studies have suggested that the release of mtDNA from mitochondria to the cytoplasm can occur through gasdermin D-pore forming complexes in the mitochondrial membrane [126], Bax/Bak-induced OMM permeabilization [127], or mPTP in the IMM [117,118]. For example, permeabilization of the IMM induced mtDNA release during apoptosis and, accordingly, deletion of Bax/Bak prevented the mtDNA release and interferon upregulation in murine ECs [127]. Although the mechanism by which mtDNA is released is still under debate, it certainly involves the permeabilization of the OMM and IMM. Importantly, these studies confirm that the release of mtDNA triggers an inflammatory response through the cGAS-STING pathway. Although this pathway has been studied in ECs in various pathological contexts [118,125,126], its role in cerebrovascular inflammation and amyloidosis is still to be defined.

In addition to its role in the activation of the cGAS-STING pathway, mtDNA has been implicated in NLRP3 inflammasome activation. It was found that NLRP3 is required for mPTP formation and release of mtDNA into the cytosol in BMDMs [99]. Several studies have indirectly linked ROS and mitochondrial dysfunction to the release of mtDNA and activation of the NLRP3 inflammasome. In human umbilical vein ECs, mtDNA-related NLRP3 inflammasome activation was observed in the presence of ROS [128]. Likewise, oxidized mtDNA bound to and activated NLRP3 in the presence of ATP-induced mitochondrial dysfunction in macrophages [110]. Interestingly, BMDMs lacking the mitochondrial transcription factor TFAM (transcription factor A, mitochondrial), which results in reduced mtDNA content, were resistant to NLRP3 activators as observed by the reduced mtROS production and lack of caspase-1 and IL-1β activation [85]. Therefore, increasing evidence suggests that mtDNA could induce NLRP3 inflammasome. However, whether this activation is a direct result of mtDNA release into the cytosol or it is mediated by mtROS release, a known activator of NLRP3, remains to be clarified.

TLR9 is a nucleotide-sensing receptor that is localized in endolysosome compartments and recognizes unmethylated cytidine-phosphate-guanosine (CpG) oligonucleotides, a characteristic that is common in DNA from bacterial origin and mtDNA. The activation of TLR9 results in the downstream activation of NFκB and the expression of pro-inflammatory cytokines [122]. Interestingly, artificial stimulation of TLR9 was able to reduce vascular plaque burden and improve cognition in Tg2576 and 3xTg animal AD models [129,130]. These data suggest that transient activation of TLR9 may induce a beneficial inflammatory response, however, the threshold for the pathological consequences of mtDNA activation of TLR9 is still unclear. Although TLR9 is expressed in ECs [131,132], to our knowledge the effect of mtDNA on TLR9 in cerebral ECs has not been investigated.

Altogether, a large body of studies demonstrate that different mitochondrial components act as DAMPs and trigger an inflammatory response by activating various pathways such as the NLRP3 inflammasome, the cGAS-STING pathway, and TLR9. The most studied mitochondrial DAMPs are the release of mtROS and mtDNA associated with mPTP opening and other mitochondrial alterations. In addition, cardiolipin, a signature phospholipid of mitochondria that is solely localized in the IMM, has been shown to activate the NLRP3 inflammasome [133]. Overall, mitochondria have been recognized as a central sensor and key mediator of inflammation and cell death that may play a critical role in the pathogenesis of inflammation in cerebral ECs during AD and CAA.

## 7. Amyloid Induces Inflammation at the Vascular Endothelium

Although it is unclear if vascular inflammation and dysfunction occur before or after microglial activation and neurodegeneration, it is likely that vascular inflammation and dysfunction drive and perpetuate neurodegeneration during CAA. One of the earliest pathological findings of human AD brains is the loss of the BBB integrity [33,134,135,136,137], which has been confirmed through in vivo and in vitro studies [28,138,139]. Early BBB permeability could be a consequence of oligomeric Aβ toxicity on ECs and its effects on mitochondrial function [27,28,140]. Mitochondrial DAMPs induced by Aβ or other vascular risk factors may activate inflammatory pathways that lead to vascular inflammation, immune cell recruitment, increased BBB permeability, and NVU dysfunction (Figure 2). This process may then precipitate neuroinflammation and neurodegeneration, important hallmarks of advanced AD.

Cell adhesion molecules expressed and released by activated ECs are involved in the recruitment of immune cells during inflammation [141]. Specifically, in AD patients, the levels of ICAM-1, VCAM-1, and E-selectin are higher in comparison to aged-matched controls [142,143] and, in particular, VCAM-1 closely associates with deficiencies in short-term memory, spatial function, and white matter changes, suggesting that it may have important biomarker capabilities [144]. Animal models of amyloidosis recapitulate the results of studies obtained in AD patients with increased levels of P-selectin, E-selectin, VCAM-1, and ICAM-1 [24]. The expression of cell adhesion molecules VCAM-1, ICAM-1, E-selectin, and P-selectin in AD may be mediated by Aβ, which was shown to promote their induction in ECs without affecting cell viability [24]. The intracellular mechanism by which Aβ accomplishes this may rely on its effect on the mitochondria and the induction of mtROS [145,146].

As discussed in previous sections, a consequence of the Aβ–RAGE interaction is the induction of mtROS. In human aortic ECs, activation of RAGE by the AGEs causes the induction of VCAM-1 through NFκB [95]. Therefore, the cellular effects of the Aβ–RAGE binding, such as the upregulation of mtROS [94], may stimulate the expression of CAMs. Other promoters of inflammation include the vascular endothelial growth factors (VEGFs) [147], which are able to stimulate the expression of CAMs through NFκB [148]. Indeed, sunitinib, an inhibitor of tyrosine kinase receptors such as VEGFR, improved memory and learning in animal models of amyloidosis [15]. Overall, these studies highlight the role of VEGF and RAGE in EC activation.

In addition to ROS, mtDNA serves as a mediator of EC inflammatory activation. In primary rat heart ECs, stimulation with dsDNA such as that from dying cells activated NFκB-dependent induction of cell adhesion molecules through the actions of TNFα [149], effects that, as discussed above, may be mediated at least partially through mtDNA [118]. Indeed, TNFα induced the expression of VCAM1, ICAM1, E-selectin, and chemokines in HUVEC cells [150]. In conclusion, Aβ induces EC dysfunction and the release of mitochondrial DAMPs such as ROS and mtDNA, which may lead to the activation of inflammatory pathways and the subsequent expression of CAMs.

Importantly, the expression of the CAMs in cerebral ECs of AD brains facilitates the recruitment of inflammatory cells. Expression of VCAM-1 in cerebral ECs leads to lymphocyte infiltration into the brain parenchyma [151]. The recruitment of these immune cells may be mediated, at least in part, by the binding of very late antigen-4 (VLA-4) receptors on lymphocytes to VCAM-1 on ECs [152,153]. In rat brain ECs, ICAM-1 cross-linking has been shown to induce Ca^2+^ signaling via PKC pathway, which is required for lymphocyte migration through the BBB [154]. Additionally, RAGE was proposed as an EC adhesion receptor and leukocyte recruiter [155]. On the other hand, neutrophils can adhere to the endothelium through the expression of lymphocyte function-associated antigen 1 (LFA-1). Aβ42 induces neutrophil activation, in association with ROS generation [24], and enhances the binding of neutrophils to ECs by inducing the high affinity state conformation of LFA-1, a conformational change that dramatically increases its affinity for ICAM [24,156]. Interestingly, the removal of LFA-1 prevents neutrophil infiltration and cognitive decline, suggesting that vascular damage and inflammation may be one of the earliest steps of AD progression [24]. The infiltration of neutrophils also triggers the release of IL-17 [24], which increases BBB permeability [157,158]. Understanding the effects of Aβ on EC-mediated immune cell migration into the brain parenchyma is important, as the infiltration of immune cells is correlated with the onset of cognitive impairment.

Increased BBB permeability is observed in CAA and AD brains [135,159,160,161,162] and is a widely known effect of Aβ on the cerebral vasculature [28,93,163]. In mouse brain ECs, Aβ42 binding to RAGE resulted in the downregulation of the TJ proteins occludin and ZO-1, and increased EC barrier permeability [164]. In addition to the decreased expression of TJ proteins, which has been observed in AD [28,165], expression of matrix metalloproteinases (MMPs) has also been associated with increased BBB permeability [137,159,166,167]. Treatment of brain ECs with Aβ leads to an increase in BBB permeability associated with enhanced expression of MMP2 and MMP9 [15,93,168], results that were also observed in the Tg2576 and 3xTg mouse models of AD [15]. In summary, high BBB permeability as a result of Aβ-induced vascular toxicity and inflammation allows the passage of peripheral immune cells and blood components into the brain parenchyma, where they are involved in the activation of microglial cells, the resident immune cells of the brain [24,169,170], and in the stimulation of multiple proinflammatory pathways.

## 8. Vascular Inflammation, Mitochondria, and Neuroinflammation

Microglial activation is mediated, at least in part, through the release of pro-inflammatory cytokines by invading neutrophils, as mentioned above [24]. However, the mitochondria are also important regulators of microglia activation. Extracellular CytC, which is normally released by dying cells, stimulates release of NO, an inflammation mediator, in murine microglial cells [26,171]. Additionally, extracellular TFAM in combination with IFN-γ augments the secretion of IL-6 in primary human microglial cells [26,172]. In the post-mortem human AD hippocampus, mitochondrial morphology is altered, indicative of accumulated mitochondrial damage [66]. Damaged mitochondria are physiologically eliminated through the process of mitophagy (mitochondrial autophagy). In animal models of AD, mitophagy was decreased in over 50% of microglia and this was associated with an increase in the number of defective mitochondria [66]. Defects in mitophagy may lead to inflammation due to the release of mitochondrial DAMPs to the extracellular space or into the cytoplasm, resulting in cellular damage [99,173,174]. Consequently, induction of mitophagy in AD microglia increases the efficiency of Aβ phagocytosis, reducing Aβ accumulation and prevented cognitive impairment in the APP/PS1 mice [66], probably due to the re-establishment of proper energy supply, as a result of the removal of damaged mitochondria.

The process of mitophagy is coupled to mitochondrial dynamics, as mitochondrial fission appears to be a requirement for mitophagy [175,176,177]. Mitochondrial fission often results in dysfunctional and fragmented mitochondria and the balance between fission and fusion is regulated by fusion proteins, including mitofusin 1 (Mfn1) and 2 (Mfn2), and optic atrophy 1 (OPA-1), and fission proteins, such as dynamin-1-like protein (DRP-1) and mitochondrial fission 1 protein (Fis1). In cynomolgus macaques, the expression of mitochondrial fusion proteins (Mfn1, Mfn2, and OPA-1) was reduced and the fission protein DRP-1 was increased after ventricular infusion of oligomeric Aβ, indicating disruption of mitochondrial dynamics and induction of fission [178]. We have recently shown that Ca^2+^-induced mPTP opening in vitro enhances proteolytic cleavage of L-OPA1 leading to accumulation of S-OPA1 [179], indicating a crosstalk between mPTP opening and mitochondrial dynamics. Based on these data, the mPTP-mediated inflammatory response in cerebral ECs may also be mediated through the inactivation of OPA1, which would increase the number of fragmented mitochondria. Importantly, stimulation of mitophagy, which potentially eliminates damaged mitochondria, was able to reduce NLRP3/caspase-1 mediated neuroinflammation in the APP/PS1 cortex [66]. Since mitophagy stimulation can abrogate the activation of inflammation, Aβ-induced mtROS per se, released to the cytoplasm, may act as the primary DAMP and trigger NLRP3 activation. Indeed, protective effects of mitophagy are mediated through the downregulation of mtROS by maintaining a healthy mitochondrial pool. In favor of this hypothesis, IL-10, an anti-inflammatory cytokine, protected against LPS-induced mitochondrial dysfunction, stimulated mitophagy, and prevented NLRP3 inflammasome activation induced by mtROS in BMDM [66,180]. Overall, the positive effects of mitophagy stimulation in APP/PS1 mice portray mitochondrial dysfunction as a causative event towards AD pathology.

Prolonged activation of microglia is common around Aβ plaques [181], where these cells are involved in the phagocytosis of Aβ and internalization to lysosomal compartments. Internalization of Aβ induces lysosomal swelling and dysfunction, and the release of cathepsin B, a cysteine protease located in the endolysosomal compartment, into the cytoplasm. Cytoplasmic cathepsin B has been proposed as the culprit of NLRP3 activation [181] and was previously detected in plaque-associated microglia [182]. It has also been shown to induce mitochondrial dysfunction through its degradation of TFAM resulting in mtROS production, a key NLRP3 activator [183,184]. Cathepsin B has also been implicated in the release of CytC from mitochondria [185] and ferroptosis [186]. Therefore, it is possible that the activation of NLRP3, as well as other inflammatory pathways, may be mediated by a variety of intracellular stress signals including cathepsin B-mediated mitochondrial damage and the subsequent release of mitochondrial DAMPs. Additionally, the defective lysosomes that secrete cathepsin B due to Aβ internalization may also result in the accumulation of damaged mitochondria due to inhibition of mitophagy. Interestingly, in microglia that lacked the NLRP3 inflammasome, the M2 phenotype predominated, leading to a decrease in Aβ deposition and improvement in spatial memory in APP/PS1 mice [65]. Additionally, the release of ASC specks from microglia has been shown to increase amyloid aggregation [187] suggesting that the NLRP3 is not only involved in the activation of inflammation, but also plays a role in Aβ deposition.

Activation of microglia also has detrimental effects on the brain vasculature perpetuating the already present damage. IL-1β released from activated microglia increased BBB permeability, downregulated the expression of TJ proteins (ZO-1, occludin, and claudin-5), and suppressed sonic hedgehog production from astrocytes, thereby reducing their protection towards BBB integrity [188]. Furthermore, the release of IL-1β from microglia also increased astrocytic activation leading to the production of pro-inflammatory cytokines C-C Motif Chemokine Ligand 2 (CCL2), CCL20, and C-X-C motif chemokine ligand 2 (CXCL2) that induce cell migration and exacerbate BBB disruption and neuroinflammation [188]. Interestingly, LPS-activated microglia increased EC barrier permeability and reduced the expression of ZO-1, occludin, and claudin-5 [189]. In conclusion, these studies demonstrate that the EC activation and BBB damage that lead to the activation of microglia are perpetuated by the further release of pro-inflammatory mediators from the microglia towards the brain vasculature.

Systemic inflammation leads to microglial recruitment to blood vessels through the chemokine receptor type 5 (CCR5), led by the release of the chemokine CCL5 by brain ECs. Microglia then start expressing claudin-5, which helps in the retention of BBB tightness [190]. Hence, the movement of glial cells [190,191] towards an activated vasculature and their upregulation of claudin and occludin proteins appears to be an initial protective mechanism to prevent worsening of vascular function. Indeed, transient activation of TLR9 was shown to decrease parenchymal and vascular Aβ pathology and improve cognitive function in various AD mouse models and in squirrel monkeys [130,192,193], suggesting that the initial activation of inflammation may be beneficial. However, a sustained inflammatory state may lead to a CD68 phagocytic phenotype and disruption of the BBB [190]. Similarly, sustained elevated levels of IL-17 induce EC dysfunction in mice fed with high salt diet, characterized by inhibition of eNOS and cognitive dysfunction [157]. Expression of RAGE in microglia of transgenic mice expressing a mutant human APP, induced IL-1β and TNFα production [92]. TNFα, in turn, has been shown to reduce the expression of claudin-5 in a NFκB-dependent manner in brain ECs [194]. Another study demonstrated that the expression of RAGE in microglia in AD subjects was higher compared to non-demented individuals [195]. A recent study using an endothelial and astrocyte co-culture demonstrated that Aβ42 counteracts the increase of CAMs in the presence of inflammatory mediators, inhibiting PBMC migration. However, the observed effects of Aβ were only present in the co-culture model, suggesting that astrocytes protect the BBB from immune cell extravasation [196]. In conclusion, vascular inflammation is an early event in CAA and AD and appears to influence neuroinflammation and progression of neurodegeneration. The mitochondria are portrayed as a central regulator of vascular activation and neuroinflammation. As a result, therapeutic strategies that prevent mitochondrial dysfunction may be also effective to reduce neuroinflammation and neurodegeneration.

## 9. Therapeutic Strategies

An increasing number of studies have centered on the mitochondrion as an initiator, sensor, and even messenger of danger signals. In this review, we portray the EC mitochondria as important initiators of Aβ-induced vascular inflammation. Therefore, therapies that prevent mitochondrial dysfunction and/or neutralize the release of mitochondrial DAMPs might prevent the activation of vascular inflammation and neurodegeneration. The most well-known mitochondrial DAMPs are mtROS, therefore therapies that prevent mtROS generation or neutralize it (such as antioxidants) could be explored as an attractive target to prevent the activation of endothelial cells in the context of CAA and AD.

Multiple mitochondria-targeted antioxidant peptides have been developed, one of which is mitoquinone (MitoQ). MitoQ is a cationic ubiquinone derivative antioxidant targeted to the mitochondria in a membrane-potential-dependent manner [197]. In microglia, MitoQ was shown to prevented mtROS and NLRP3 activation in a model of intracerebral hemorrhage [198]. Similarly, in pulmonary aortic EC, MitoQ reduced TNFα induced expression of ICAM-1 and NFκB [199]. However, further studies demonstrated that higher concentrations of MitoQ could induce ICAM-1 expression through the induction of ROS production [199,200], carbonylation, and glutathionylation of cellular proteins [199]. Therefore, the potential benefits of MitoQ as powerful mitochondria-targeted antioxidant are controversial.

In addition to MitoQ, another peptide was developed called the Szeto-Schiller-31 (SS-31) peptide. The peptide SS-31 is a mitochondria-targeted antioxidant that mostly accumulates in the IMM, the major site for mtROS production, and can cross the BBB [201,202]. Treatment of aged WT mice with SS-31 improved CBF, neurovascular coupling, and spatial learning [203]. In addition, in primary cerebrovascular ECs from aged rats, SS-31 improved mitochondrial respiration and reduced mtROS production [203], suggesting that SS-31 could potentially inhibit the mtROS-induced inflammatory response. Indeed, SS-31 was able to reduce LPS-induced oxidative stress and inflammation in murine microglia [204], and in the mouse hippocampus it prevented mitochondrial dysfunction, oxidative stress, inflammation, memory impairments, and neuronal death in the presence of LPS [205]. Although SS-31 has been studied in the context of AD with positive results (reviewed in [202]), the potential role of SS-31 in preventing cerebrovascular EC activation in models of CAA or AD is still unclear.

In addition to mtROS, inhibition of the mPTP, which can be activated by mtROS [115,116], would be an attractive strategy to prevent Aβ-induced mitochondrial dysfunction and inflammation. The formation of the mPTP has been linked to mtROS and the release of mtDNA into the cytoplasm [117,118]. As a result, mPTP inhibition would prevent the release of mitochondrial DAMPs and the activation of the downstream inflammatory pathways. Several studies have demonstrated that drugs that inhibit CypD, such as SfA and CsA, are able to prevent the activation of inflammation. CsA was demonstrated to prevent LPS-induced caspase-1 activation and IL-1β secretion in BMDM [99,109,110] and was able to prevent astrocytic reactivity in AD models [206], albeit through calcineurin inhibition [111]. Interestingly, in human aortic ECs SfA inhibited mtROS production and improved vascular relaxation in a model of hypertension [207]. However, it was reported that *CypD^−/−^* macrophages still exhibited NLRP3 activation [109], raising the question of whether these drugs are sufficient to inhibit mPTP formation or if the mPTP is at all involved in the release of mitochondrial DAMPs into the cytoplasm. As a result, further studies are required to assess the potential of CsA and SfA as inhibitors of inflammation. One of the most challenging aspects of mPTP inhibition is the lack of knowledge on the molecular identity of the mPTP, making the development of specific pharmacological inhibitors impossible. Taking into consideration that the mPTP is involved in other diseases with an inflammatory component [101,102,103,104,105,106,107], studies aiming at elucidating the molecular identity of the mPTP and the development of specific mPTP inhibitors would be of great interest to the scientific community.

Finally, recent evidence has demonstrated that carbonic anhydrase inhibitors (CAi), a family of FDA-approved, BBB-permeable drugs, may have protective roles on the mitochondria. Cytochrome c is normally located in the outer part of the IMM, and its release into the cytosol is indicative of mitochondrial damage and OMM rupture. The CAi Methazolamide (MTZ) was found to inhibit CytC release out of a list of compounds from the NINDS library of neurodegeneration drug screening consortium [208]. It was shown that MTZ prevents Aβ-induced mtROS production, CytC release, and caspase activation in neuronal and glial cells in culture and prevented neurodegeneration in mice after intra-hippocampal Aβ injection [209]. Furthermore, both MTZ and Acetazolamide (ATZ), another clinically used CAi, tested in cells of the neurovascular unit challenged with Aβ, prevented loss of mitochondrial membrane potential and mtROS production [27]. For an in-depth review on the therapeutic potential of CAi in the context of CAA and AD, the reader is directed elsewhere [210]. Although the role of MTZ and ATZ in EC activation during CAA and AD has not been fully investigated, it is conceivable that the protective effect they have on mitochondrial function might translate into decreased EC activation in the presence of Aβ.

In conclusion, drugs that prevent mitochondrial dysfunction or provide antioxidant capabilities could provide protection against EC activation, and possibly EC death. However, it is important to also consider other risk factors that have been associated with AD, such as APOE status. The presence of the APOE4 allele dramatically increases the risk for developing AD with age, decreases the age of onset [211,212], and is considered the major genetic risk factor for late-onset AD. Individuals with APOE4 (ε3/ε4 and ε4/ε4 alleles) develop increased BBB permeability in the hippocampus and medial temporal lobe, and cognitive decline independent of their Aβ and Tau pathology [213]. Furthermore, APOE4 has also been linked to mitochondrial dysfunction [214,215,216]. Therefore, it is critical to evaluate how risk factors, such as APOE status, might add a level of complexity to mitochondria-targeted therapies.

## 10. Conclusions

Overall, current knowledge suggests that amyloid and vascular risk factors can cause mitochondrial dysfunction and ROS production in cerebral ECs, thereby stimulating vascular inflammation. This vascular activation induces immune cell recruitment, with microglial and astrocytic activation. The resulting widespread inflammatory cascade, in turn, has been shown to precipitate endothelial and neurovascular mitochondrial damage, eliciting a detrimental impact on the BBB. As a result, a progressive worsening of vascular function and neuroinflammation, loss of BBB integrity, and impaired Aβ clearance will further perpetuate neurodegeneration, accelerating the development and progression of dementia and AD. Understanding the mechanisms involved in the pathogenesis of neural and vascular inflammation during AD and CAA will allow the development of new therapeutic strategies for the treatment of patients with AD and other neurodegenerative diseases. A growing number of studies suggest a potential role of the mitochondria as a nexus of stress in mediating inflammatory signaling in cerebral ECs and other neurovascular cells, underlining the importance of further studies in this area.

## Figures and Tables

**Figure 1 cells-10-02903-f001:**
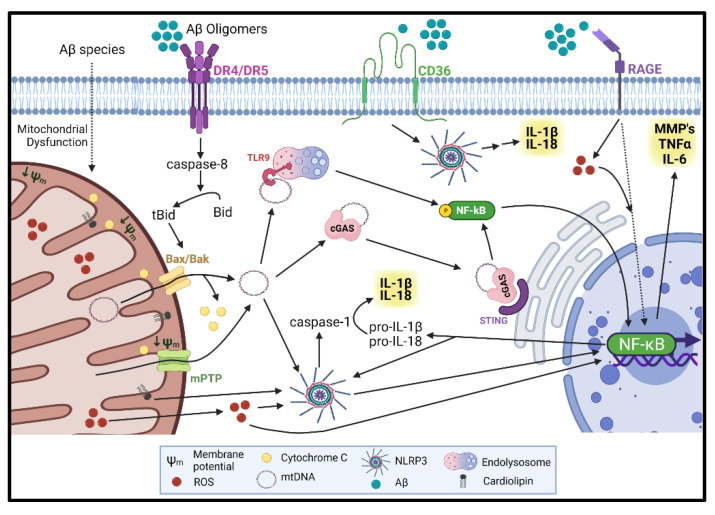
Aβ induces the release of mitochondrial DAMPs in cerebral endothelial cells (ECs) inducing vascular activation. Mitochondrial dysfunction is induced by Aβ, at least in part, through the binding and activation of TRAIL Death Receptors (DRs) and activation of the extrinsic and intrinsic apoptotic pathway. Amyloid β has been shown to induce loss of mitochondrial membrane potential (Ψm), increase in the generation of mitochondrial reactive oxygen species (mtROS), and permeabilization of mitochondrial membranes leading to the release of cytochrome C and mitochondrial DNA (mtDNA) into the cytoplasm of endothelial cells. The increase of mtROS leads to the activation of the NLRP3 inflammasome, activation of caspase-1 and NFκB, resulting in the release of IL-1β and IL-18. Permeabilization of the inner mitochondrial membrane (IMM) and outer mitochondrial membrane (OMM) lead to the release of mtDNA to the cytoplasm. The presence of double stranded DNA (dsDNA) activates the NLRP3 inflammasome, toll-like receptor 9 (TLR9) on the endolysosomal compartment, and the cGAS/STING pathway on the cytosol and endoplasmic reticulum (ER) membrane, which lead to the activation of NFκB and upregulation of more pro-inflammatory cytokines. Amyloid β can also activate the NLRP3 inflammasome by binding to the CD36 membrane receptor, or increase the production of ROS through the RAGE receptor. The exponential increase in pro-inflammatory cytokines leads to blood–brain barrier (BBB) disruption, through activation of matrix metalloproteinases (MMPs), downregulation of tight junction (TJ) proteins, EC activation, and expression of cell adhesion molecules. Figure created with BioRender.com.

**Figure 2 cells-10-02903-f002:**
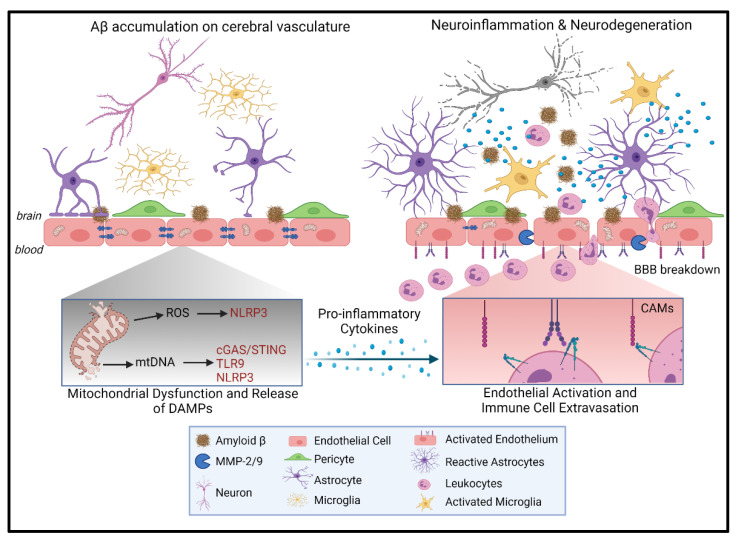
Vascular inflammation and immune cell extravasation drive neuroinflammation and neurodegeneration. The Aβ-mediated release of mitochondrial DAMPs, such as mitochondrial reactive oxygen species (mtROS) and mitochondrial DNA (mtDNA), from endothelial cells (EC) induces a perivascular inflammatory response. This in turn, results in EC activation, characterized by the expression of cell adhesion molecules (CAMs). These CAMs, expressed on the EC membrane, induce the recruitment of circulating immune cells through receptor binding. Immune extravasation occurs through the blood–brain barrier (BBB) due to loss of BBB integrity. Loss of BBB integrity is mediated by the matrix metalloproteinases (MMP) MMP2 and MMP9 and the downregulation of tight junction proteins. Infiltrated immune cells also secrete pro-inflammatory cytokines resulting in activation of astrocytes and microglia, which then release additional pro-inflammatory cytokines. This perpetuated immune activation leads to neurovascular cell damage and neurodegeneration. Figure created with BioRender.com.

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
