# Peer review of "Dissecting the Crosstalk between Endothelial Mitochondrial Damage, Vascular Inflammation, and Neurodegeneration in Cerebral Amyloid Angiopathy and Alzheimer’s Disease"

_cells, 2021, doi:10.3390/cells10112903_

Round 1
Reviewer 1 Report
The authors thoroughly reviewed articles regarding cerebral amyloid angiopathy resulting from Aβ deposition and Alzheimer’s disease by focusing on mitochondrial damage of vascular endothelial cells and vascular inflammation. Wide-ranged issues, particularly basic aspects, are comprehensively described.
This is an interesting article shedding light on vascular aspects of Alzheimer’s disease. It provides important insights into the pathophysiology of this disease. Taking the prevalence of this disease into consideration, this article will attract broad range of readers from basic scientists to physicians. The manuscript is well written, and I enjoyed reading it.
Although I do not have any critical comments, minor issues and suggestions to strengthen this manuscript are raised as follows:
- Cerebral amyloid angiopathy has also been reported in patients with amyloidosis resulting from proteins other than Aβ (Neurology 2005; 65: 1051-6, Neurology 2016; 87: 773-81). I would suggest including this issue.
- Before researchers pay attention to an importance of Aβ oligomers, the relationship between amyloid fibrils and vessels were investigated in detail (Acta Neuropathol 1992; 84: 117-27). This issue is important from the historical viewpoint.
- The morphology of angiopathy has also been detailed in another type of amyloidosis (Neurology 2016; 87: 2220-2229). As the comparison among different types of angiopathy will provide important insights into the pathophysiology of amyloidosis, this issue should be incorporated, by citing this study.
- The audience may feel that aducanumab was formally approved in 2016 by FDA when they read the first paragraph of the introduction section. Actually, it was approved in 2021 using the accelerated approval pathway based on the reduction of the level of amyloid plaques in the brain but not the slowing of cognitive decline. As this issue was summarized in a recent review (Molecules 2021; 26: 5091), it should be clarified by citing this article.
- A discussion on therapeutic strategies based on the mechanisms of mitochondrial damage of vascular endothelial cells and vascular inflammation will increase clinical interest of this manuscript.
Author Response
Reviewer 1
- Cerebral amyloid angiopathy has also been reported in patients with amyloidosis resulting from proteins other than Aβ (Neurology 2005; 65: 1051-6, Neurology 2016; 87: 773-81). I would suggest including this issue.
A sentence has been added to clarify that CAA is not solely present in Aβ amyloidosis (lines 53-54)
However, it is important to note that CAA is not exclusive to amyloidosis due to Aβ [11,12].
- Before researchers pay attention to an importance of Aβ oligomers, the relationship between amyloid fibrils and vessels were investigated in detail (Acta Neuropathol 1992; 84: 117-27). This issue is important from the historical viewpoint.
A sentence highlighting these findings and the reference has been added (lines 131-133)
Indeed, early studies report that in human AD brains, the accumulation of Aβ fibrils in cerebral vessels provoke EC degeneration and reduction on vessel size [59].
- The morphology of angiopathy has also been detailed in another type of amyloidosis (Neurology 2016; 87: 2220-2229). As the comparison among different types of angiopathy will provide important insights into the pathophysiology of amyloidosis, this issue should be incorporated, by citing this study.
This reference has been added (lines 51-53).
The vascular accumulation of Aβ occurs mostly in cortical and leptomeningeal arteries and capillaries and is often associated with cerebral microhemorrhages, increase in blood-brain barrier (BBB) permeability, and inflammation [9,10].
- The audience may feel that aducanumab was formally approved in 2016 by FDA when they read the first paragraph of the introduction section. Actually, it was approved in 2021 using the accelerated approval pathway based on the reduction of the level of amyloid plaques in the brain but not the slowing of cognitive decline. As this issue was summarized in a recent review (Molecules 2021; 26: 5091), it should be clarified by citing this article.
Thank you, this mistake was fixed to read 2021 and the article has been cited (lines 35-39).
In 2021, after nearly twenty years of attempts to develop new therapeutic strategies, aducanumab, a human monoclonal antibody that selectively targets aggregated amyloid-β (Aβ), has been approved (through the accelerated approval pathway) by the FDA for treatment of patients with AD [2]. However, the therapeutic effectiveness of aducanumab is still debated by many research groups [2-4].
- A discussion on therapeutic strategies based on the mechanisms of mitochondrial damage of vascular endothelial cells and vascular inflammation will increase clinical interest of this manuscript.
Thank you, the authors agree with the reviewer and have added a final section on therapeutic strategies to the manuscript (lines 628-707).
Reviewer 2 Report
The authors have provided a thorough review of the literature surrounding endothelial cell dysfunction in Alzheimer's Disease, with a particular focus on inflammation and the role of mitochondria. As the authors have discussed a number of processes and pathways the inclusion of additional figures and/or tables may improve the clarity of the information provided.
Minor comments:
- Page 4 lines 186-188, please provide references for this statement.
- Symbols are missing from figure 1 legend.
Author Response
Reviewer 2:
- Page 4 lines 186-188, please provide references for this statement.
The references have been added (lines 190-192)
Due to their important role in brain cell health, alterations of mitochondrial structure and function in cerebrovascular ECs induce cell dysfunction, loss of BBB integrity, and inflammation, eventually leading to cell death [27,39,73-75].
- Symbols are missing from figure 1 legend.
Thank you, we have added all missing abbreviations.
For Figure 1 (lines 239-243 and 246-247).
For Figure 2 (lines 443-444).